# Bioprinting Technology in Skin, Heart, Pancreas and Cartilage Tissues: Progress and Challenges in Clinical Practice

**DOI:** 10.3390/ijerph182010806

**Published:** 2021-10-14

**Authors:** Eleonora Di Piazza, Elisabetta Pandolfi, Ilaria Cacciotti, Andrea Del Fattore, Alberto Eugenio Tozzi, Aurelio Secinaro, Luca Borro

**Affiliations:** 1Multifactorial and Complex Disease Research Area, Preventive and Predictive Medicine Unit, Bambino Gesù Children’s Hospital, IRCCS, 00146 Rome, Italy; eleodp@gmail.com (E.D.P.); albertoeugenio.tozzi@opbg.net (A.E.T.); 2Engineering Department, Niccolò Cusano University of Rome, INSTM RU, 00166 Rome, Italy; ilaria.cacciotti@unicusano.it; 3Genetics and Rare Diseases Research Area, Bone Physiopathology Research Unit, Bambino Gesù Children’s Hospital, IRCCS, 00146 Rome, Italy; andrea.delfattore@opbg.net; 4Clinical Management and Technological Innovations Area, Department of Imaging, Bambino Gesù Children’s Hospital, IRCCS, 00146 Rome, Italy; aurelio.secinaro@opbg.net (A.S.); luca.borro@opbg.net (L.B.)

**Keywords:** bioprinting, skin, heart, pancreas, cartilage, tissue engineering, regenerative medicine, clinical applications

## Abstract

Bioprinting is an emerging additive manufacturing technique which shows an outstanding potential for shaping customized functional substitutes for tissue engineering. Its introduction into the clinical space in order to replace injured organs could ideally overcome the limitations faced with allografts. Presently, even though there have been years of prolific research in the field, there is a wide gap to bridge in order to bring bioprinting from “bench to bedside”. This is due to the fact that bioprinted designs have not yet reached the complexity required for clinical use, nor have clear GMP (good manufacturing practices) rules or precise regulatory guidelines been established. This review provides an overview of some of the most recent and remarkable achievements for skin, heart, pancreas and cartilage bioprinting breakthroughs while highlighting the critical shortcomings for each tissue type which is keeping this technique from becoming widespread reality.

## 1. Introduction 

Three-dimensional bioprinting is an additive manufacturing technology, which applies the principles of 3D printing to the biomedical field. Its goal is to create in vitro viable and functional biological constructs with complex architectures which are able to mimic the native organs, and to provide powerful platforms for studying tissue development and homeostasis and for modeling diseases in pharmaceutical testing [1].

The bioprinter shapes the tissue through a layer-by-layer controlled deposition of a bioink consisting of growth factors (optional), a peculiar cell type and a biomaterial fluid matrix in which cells are embedded and interspersed. Thanks to its interdisciplinary nature, the main areas of bioprinting application involve drug testing, disease modeling and tissue engineering [2,3,4]. Despite the critical issues that currently preclude an in vivo application, in vitro personalized disease modeling and drug screening remain the closest reality to clinical application.

Bioprinted tissues have not yet reached the structural and functional organicity necessary to make them a valid substitute for allografts, despite 10 years of research and progress on this topic. Yet while a routine clinical application of bioprinted materials may be rather far off, many important advancements have been made in this field. The greatest challenges lay in the complexity of reproducing a viable tissue with all its biological features which is able to integrate within the host tissue and to mechanically and physiologically support the self-renewal of the damaged organ. For these reasons, an ideally engineered bioprinted design should have a degradation rate that fits the timing for endogenous regenerative processes, simultaneously undergoing vascularization and innervation [5]. In addition, ethical questions and proper regulatory aspects should be considered [6]. In this review, we provide an overview of the latest major achievements in bioprinting tissue prototypes and discuss recent developments, current challenges and the future prospects for the design and realization of 3D bioprinting for complex tissues such as skin, heart, pancreas and cartilage.

## 2. Methods

We conducted a scoping review following the method of Arksey and O’Malley (2005): (i) identifying the research question and the relevant studies; (ii) selecting studies; (iii) summarizing the data and reporting the results.

### 2.1. Identifying the Research Question

We identified three research questions: (i) What evidence exists on bioprinting as to clinical aspects? (ii) What are the limitations of this new technique? (iii) What are the future challenges?

### 2.2. Identifying Relevant Studies

The present scoping review identified, retrieved and evaluated information from peer-reviewed articles that examined the impacts of bioprinting on clinical practice in regard to four different tissues: cardiac tissue, skin, cartilage and pancreatic tissue.

We focused on studies published between 1 January 2016 and 31 December 2020 and consulted three databases (PubMed, Google Scholar and Medline) using the following search strings: “bioprinting AND cardi*”, “bioprinting AND skin”, “bioprinting AND cartilage”, “bioprinting AND pancreatic*”. This period was selected because the bioprinting technique and its related applications are relatively recent and most of the bioprinting studies have been published in these years.

### 2.3. Selecting Studies

Only empirical papers with an English language abstract were included. We considered all types of research design. We applied the following exclusion criteria at two stages of the study selection: screening by title, abstract and full text.

Figure 1 summarizes the selection process. Using the specific search string, the search yielded 233 articles for cardiovascular tissue, 187 articles for skin, 26 articles for pancreatic tissue and 286 articles for cartilage. When restricted to the articles published in the past five years and written in the English language, the search yielded 101 articles for cardiovascular tissue, 68 articles for skin, 101 articles for cartilage and 16 articles for pancreatic tissue. Based on the screening of the title and the abstract and excluding the reviews, 17 articles for cardiovascular tissue, 20 articles for skin, 34 articles for cartilage and 6 articles for pancreatic tissue were included in this review.

The selected articles by type of tissue are reported in Table 1, Table 2, Table 3 and Table 4.

## 3. Main Bioprinting Techniques

Bioprinting is based on technologies used in mechanical and biomedical engineering. There are different types of bioprinting techniques based on different types of 3D printers and on the deposition system of the biomaterial on the print plate, as schematized in Figure 2, where the applications for the four tissues considered in this review paper are evidenced. The most important bioprinting techniques are extrusion, inkjet and laser printing, but other currently used techniques include microvalve-based [7] and stereolithography (SLA) printing.

### 3.1. Extrusion Printing

One of the most relevant bioprinting technologies is extrusion printing [7] that derives from FFF (fused filament fabrication) technology normally used by conventional 3D printers for printing thermoplastic materials. This type of printing is based on the use of 3D printers formed by pressurized syringes loaded with cells embedded in a bioink, and is particularly suitable for extrusion of viscous gel materials.

The main advantage of extrusion 3D printing is the possibility of extruding cells, extracellular matrices and thermo-biopolymers both individually and simultaneously with more than one extrusion system. This is a bioprinting technique widely used in the bone tissue regeneration field [8] and in drug release studies [9]. The extrusion technique also allows the printing of scaffolds with thermo-biopolymers such as PCL (polycaprolactone) [10,11], PLA (polylactic acid) [11] and PVA (polyvinyl alcohol) [11]. The disadvantage of this technology is the poor print resolution depending on the needle’s size, making it difficult to print very small objects.

### 3.2. Inkjet—Based Bioprinting

Inkjet is a technology that uses bioink microdroplets which are deposited on culture plates or various substrates. Microdroplets are deposited from a bioink-filled extrusor by positive pressure that causes the drops to escape from the printing nozzle. The pressure which causes droplets to leak can be generated by a heat source in the syringe that creates a positive compression bubble with which the bioink lets it out, or by a piezoelectric system. In the case of the thermal system, the obtained droplets are made up of different sizes and frequently cause the nozzle to clog, which results in the printing of objects that are not particularly smooth. In the case of the piezoelectric system, the cells can be damaged by lysis of the cell membrane.

The inkjet technology has been successfully used in the printing of DNA molecules [12], and of ovarian cells in hamster and rat motor neurons [13]. In 2015, heart valves were printed with this technology although their functional properties have to be still unraveled and studied [14].

One of the main criticisms of the inkjet printing is the poor mechanical consistency of the bioinks which makes the constructs poorly manipulated. It is necessary to develop bioinks that allow for the production of more solid and structurally defined biological objects.

### 3.3. Laser—Assisted Bioprinting (LAB)

Laser-assisted bioprinting (LAB) is a technique that allows printing cells and liquid materials with a micrometer resolution. Laser-assisted bioprinters consist of three components: (1) a pulsed laser source, (2) a target to which a biological material is printed, and (3) a receiving substrate that collects the printed material. The ribbon is made of a thin absorbing layer of metal (such as gold or titanium) coated onto a laser transparent support (i.e., glass). Organic material (molecules or cells) is prepared in a liquid solution (i.e., culture media), and deposited on the surface of the metal film. The laser pulse induces vaporization of the metal substrate, resulting in the production of a jet of liquid solution which is deposited onto the facing substrate. LAB is an effective tool for in situ printing of a bone substitute, due to its high printing resolution and precision [15,16].

### 3.4. Microvalve—Based Bioprinting

Microvalve-based bioprinting is a system comprised of a movable robotic platform and an array of multiple electromechanical microvalve print-heads. The bioink ejection is controlled by the pneumatic pressure applied through a gas regulator and a system of micro-valves, which offers a controlled deposition of materials through a layer-by-layer fabrication approach [7]. This technique has some main advantages: it allows the synchronized ejection of biomaterials and cells from different print-heads and the deposition of a thin material layer with a precise cellular positioning and high throughput printing. However, it prints only hydrogels within a limited range of viscosity (∼1 to 200 mPa s) and cell concentrations of up to 10^6^ cells per ml due to the clogging issues in the small nozzle orifice (100–250 μm) [7,17]. The material deposition is highly dependent on the nozzle diameter, the viscosity and surface tension of the bioink, the pneumatic pressure and the valve opening time.

### 3.5. Stereolithography (SLA)

Stereolithography (SLA) is an early and widely used 3D printing technology. In the early 1980s, a Japanese researcher invented the modern layered approach to stereolithography by using ultraviolet light to cure photosensitive polymers. Stereolithography works by focusing an ultraviolet (UV) laser onto a vat of photopolymer resin [18]. With the help of computer-aided manufacturing or computer-aided design (CAM/CAD) software, the UV laser is used to draw a pre-programmed design or shape on the surface of the photopolymer vat. Photopolymers are sensitive to ultraviolet light, so the resin is photochemically solidified and forms a single layer of the desired 3D object. The liquid materials used for SLA printing are commonly referred to as “resins” and are thermoset polymers. Stereolithographic models have been used in medicine since the 1990s, and they are used as an aid for diagnosis, preoperative planning as well as implant design and manufacture. An example of stereolithography’s application is in rehearsing osteotomies, but also in the production of prototypes and models to help plan surgeries. The main advantage of this technique is its speed, since some functional parts can be manufactured within a day, but the cost represents a limit to its diffusion.

## 4. 3D Bioprinting of Skin 

### 4.1. Background

Human skin is a multi-layered tissue in which a complex system of cells, glands, nerves and blood vessels cooperate to fulfill several important functions in our body [19,20]. 

Skin wounds due to trauma, ulcers, burns or other causes represent a social and economic burden, being quite common [21]. Nowadays, the gold standard clinical treatment for these injuries is skin autograft, but its application is strictly related to the extension of the wound [22]. Other common strategies include allografts, liquid formulation, wound dressing and even cell spray techniques [23,24]; more often, traditional graft approaches are combined with skin substitutes. However, despite their great benefits in clinical practice, these substitutes do not fully recapitulate the complexity of native skin, and they display some side effects according to their composition, together with a high cost of production [22]. This disadvantage may be overcome by bioprinting. 

### 4.2. Current Applications and Future Perspectives

Bioprinting represents an emerging tool that could overcome the gap between grafts and skin substitutes. 

In a notable study, Cubo et al. bioprinted human skin (dermal and epidermal layers) in less than 35 min, making an important step towards the needs of clinical reality (Table 1). The authors bioprinted primary human fibroblasts (hFBs) within a plasma-derived hydrogel and then seeded primary keratinocytes: clear stratification and differentiation of dermis and epidermis occurred, as shown by the presence of rete ridges (peculiar to human skin) in some regions [25].

Pourchet et al. developed a unique bioink combination of alginate, fibrinogen and gelatin with hFBs for dermal matrix, and then layered keratinocytes on the top (Table 1). A complete evaluation of bioprinted skin morphology was performed: epithelium and dermis maturation, ECM production and the formation of desmosomes and hemidesmosomes demonstrated structural similarity to human normal skin [26]. 

In the work of Kim et al., the skin substitute was realized using a novel hybrid printing system (integrated composite tissue/organ building system) in which extrusion and inkjet printing are combined in a single-step process (Table 1). A polycaprolactone (PCL) mesh supported the printing of the dermal layer (collagen + fibroblasts), preventing collagen shrinkage; then, the inkjet printer uniformly seeded keratinocytes on the dermal matrix. After 14 days, 3D printed layers displayed a good degree of maturation, as shown by the cellular spatial distribution and the expression of dermal and epidermal markers [27]. One year later, the authors tested the properties of porcine decellularized extracellular matrix (dECM), processing it and using it as a bioink. The dECM was demonstrated to mostly retain its composition after processing and also proved to be a biomimicking substrate, setting the proper conditions for cell growth and differentiation. Moreover, the authors succeeded in bioprinting a pre-vascularized skin patch (made of adipose-derived stem cells and endothelial progenitor cells in dECM), which resulted in a significantly improved neovascularization and re-epithelialization after a graft in an athymic nude mouse model [28]. This represents a remarkable issue for engineered constructs, since they must go through a perfect integration with the host endogenous tissues, in addition to being vascularized and innervated. Other approaches to induce angiogenesis in tissue remodeling involve the fabrication of scaffolds soaked with growth factors, or absorbing pro-angiogenic molecules within the bioink [29]. 

Melanocytes reside in the basal layer of the epidermis and they shield skin surface from UV radiation through melanin production, thus playing an important protective role [30]. Some authors tried to produce skin models including also this cell type, with different outcomes [31,32,33] (Table 1). Recently, to overcome poor printability and long cross-linking duration of collagen biomaterials, Ng et al. reported the use of polyelectrolyte gelatin-chitosan (PGC) hydrogel for 3D bioprinting of skin. They optimized a co-culture medium for human melanocytes, keratinocytes and hFBs, and they bioprinted a uniformly pigmented skin, with human melanocytes anchored at the basement membrane and well stratified and mature layers of dermis and epidermis [34] (Table 1). This could represent a promising starting point for further improvements in co-culture techniques.

All these latest remarkable achievements bring us closer everyday to the clinical application of bioprinting in the treatment of skin defects, but currently this is not yet a therapeutic option since there are several key points that represent critical limitations to overcome.

First of all, bioprinted skin should have the proper thickness, texture and permeability, and the biological construct should achieve the right internal porosity to grow and integrate in vivo [35,36]. Unfortunately, the proper permeability is often not reached in the printed constructs [24]. On the other hand, an essential requirement concerns vascularization and innervation: indeed, these goals are far from being fully achieved for bioprinted skin, despite the progress in the field [3].

Another critical issue is the integration within the construct of skin appendages and different cell types: no current bioprinted prototype of human skin fully recapitulates all cells and appendages present in native skin [24], even if there is active research in the field. Finally, the importance of GMP (good manufacturing practices) requirements should be considered: all the cells, the biomaterials and the products destined for clinical use should undergo prior validation of the production process, quality assessments and regulatory approval [4,24,37]. Furthermore, the intrinsic time- and cost-effectiveness of such technology have an impact on its everyday use in clinical practice.

## 5. 3D Bioprinting in Cardiovascular Disease

### 5.1. Background

Cardiovascular diseases are a leading cause of mortality, affecting a lot of people worldwide [38]. In China and the USA, cardiovascular diseases are recognized as a major national health concern; their burden grows over time and it [39,40] is estimated that one in three deaths are due to cardiovascular diseases.

Myocardium is a very complex tissue, with a hierarchical structure [41], made of a thick weave of highly specialized cells (cardiomyocytes, CMs), fibroblasts (FBs) and blood vessels. It has poor regenerative capacities and an electrical activity that must be well coordinated and responsive to physiological and pharmacological stimuli; one of the main risks of implanting a cardiac patch is the unsuccessful integration with the host cardiac frequency, thus generating arrhythmias [42,43]. A myocardial injury (such as an infarction) causes CMs death and the formation of scarring tissue, with possible heart failure in the case of extensive damage [44]. Currently, the main therapeutic option for severe heart failure is organ transplantation, but it presents two major limitations: the lack/scarcity of donors and the possible rejection by the immune system [45]. Thus, a promising future option could be represented by the tissue engineering approach, associated with the bioprinting technique [46].

Notably, Lee et al. bioprinted a contractile left ventricle model seeded with human stem cell-derived cardiomyocytes (hESC-CMs) and cardiac fibroblastsin the core region, using a novel bioprinting procedure (“FRESH—Freeform Reversible Embedding of Suspended Hydrogels system) (Table 2). They also bioprinted a tricuspid heart-valve and a heart of neonatal size, achieving scalability, high printing reliability and accuracy [47]. In a recent study, Noor et al. bioprinted a vascularized cardiac patch and a mini heart-like structure using a “personalized” bioink, made of a hydrogel derived from human omental tissue and of CMs + endothelial cells differentiated from human iPSCs (Table 2). The patch showed sarcomeric organization, contractile potential and also a prototype of a vascular network [48]. Kupfer et al. also bioprinted iPSCs-derived CMs and endothelial cells in a chambered mini-sized heart, endowed with thick-walls and electromechanical function [49] (Table 2); this could represent a promising model for in vitro studies. Similarly, in another study, a bioprinted cardiac patch physiologically responded to cardiac drugs (carbachol and epinephrine), reversibly modulating the beating frequency, thus showing potential relevance in drug testing [50]. 

Among the most used bioinks in the cardiac field are dECM and its components [51,52,53]; several new compositions have been/are being investigated [54] and even scaffold-free approaches, involving the bioprinting of cellular spheroids on a micro-needles platform as a support [55]. An interesting example of this scaffold-free technology is described by Ong et al., who co-cultured iPSC-CMs, FBs and endothelial cells in different ratios to form cardiospheres (Table 2). Then cardiospheres were selectively picked up by a robotic arm and very precisely bioprinted on a needle array. Spheroid fusion formed cardiac patches electrically well integrated and able to engraft in vivo, but they showed poor mechanical properties and arrhythmias were detected in the regions at higher FBs concentrations [56]. In vivo analysis was performed in a successive work, with grafted patches increasing survival rate in infarcted rats and slightly improving cardiac function [57]. However, the short follow-up period and a lack of coordination between endogenous and iPSCs-CMs contraction rate have to be considered in this study [57].

### 5.2. Current Applications and Future Perspectives

Three-dimensional stem cell bioprinting approaches can have huge implications in regenerative medicine, for the modeling and treatment of heart disease and failure. One of the most dynamic research topics of bioprinting is actually focusing on cardiac valve and myocardial regeneration, with a specific interest in creating a prototype of a biofunctional cardiac patch [58,59].

The first aspect to consider in bioprinting myocardial tissue is the cell source. Adult cardiomyocytes are difficult to expand in vitro [60], and their availability is scarce. For this reason, several alternative cell types have been considered, including MSCs (mesenchymal stem cells), human cardiac progenitor cells (hCPCs) [61], ESCs (embryonic stem cells) and iPSCs (induced pluripotent stem cells) [48,62]. However, all these cell types have some important limitations: unlike ESCs, iPSCs-derived cardiomyocytes do not trigger any immune reactions in the patient since they are endogenous cells, but on the other hand they often display an incomplete differentiation into cardiomyocytes [63]. As with stem cells, use of both iPSCs and ESCs raises concerns about safety because of their teratogenic potential [64]. Finally, the clinical use of ESCs implies some ethical questions, since they can come from the disruption of an embryo [65]. Another aspect to consider is that, independently of the cells used, a damaged microenvironment—such as the one of the host after an infarction—can impinge on the survival rate, differentiation capabilities and functionality of the transplanted cells [65].

The scaffold-free 3D bioprinting approach could be the technique which mostly mimics the native tissue physiology, but currently the bioengineering and fabrication techniques are not advanced enough to guarantee an adequate fit for clinical purposes [66].

Nowadays, there is no clinical application of bioprinting for heart repair: engineered myocardium grafts are currently at the preclinical study level and may very well represent economical and efficient solutions to myocardial infarction in the future [67,68].

Besides the limitations related to the type of cells employed in the fabrication of a patch, another critical shortcoming of all the bioprinted grafts is that the implant should be highly vascularized to allow cell survival. Indeed, the construct has to reach an in vivo degree of vascularization not limited to capillaries, but implying an organized and hierarchical network of blood vessels, able to support and nourish the implanted cells. This level of vascularization for bioprinted patches has not been fully achieved [34].

Further studies are needed for printing adequately vascularized heart tissue of clinically-relevant thickness that can appropriately respond to electrical impulses and maintain a synchronous beating pattern [68], since a synchronous and host-integrated contractility rate should be reached by the implant. To improve this aspect, several kinds of biomaterials have been optimized, such as silicon-nanowire field-effect transistors integrated with collagen, alginate and PLGA (poly(lactic-co-glycolic acid)) scaffolds that have been used to monitor the electrical activities of seeded cardiomyocytes [69]. Further efforts need to be undertaken in the use of nanoelectronic scaffolds [70] to provide electrical and mechanical stimulation to the cells in order to promote cardiomyocyte growth and stimulation. Additionally, next generation scaffolds should facilitate spontaneous beating of engineered cardiac tissue such as through the incorporation of carbon nanotubes in hydrogels [71], which has been shown to enhance viability and phenotypical features of rat ventricular myocytes [69].

Finally, some critical roadblocks—common for all bioprinted organs—have to be overcome to bring cardiac bioprinting into clinical practice: scalability, cost-effectiveness, and the establishment of GMP production rules and precise regulatory guidelines [42,72].

## 6. 3D Bioprinting of Pancreatic Tissue

### 6.1. Background

In vitro 3D models have been developed to investigate the in vivo tumor biology, microenvironment and growth conditions of pancreatic cancer cells and to identify new therapeutic approaches for diabetes [73,74]. Diabetes is a major health concern worldwide with a significant burden in terms of clinical and socioeconomic impact [75]. Its prevalence is increasing worldwide, with an estimate of approximately 1 in 300 by 18 years of age in the United States [76]. Type 1 diabetes mellitus (T1DM) is a chronic autoimmune disease caused by a dysfunction of pancreatic beta cells with a loss of insulin secretion [77]. Most beta cells are destroyed before clinical onset and this leads to persistently high levels of glucose in the blood [78]. Longstanding hyperglycemia leads to several complications including neuropathy, cardiovascular disease, nephropathy and retinopathy. T1DM management requires a high patient compliance with multiple daily blood glucose measurements and subcutaneous insulin injections [79].

New approaches providing cellular replacement strategies could radically change the management of the disease and the quality of life of diabetic patients. Much progress has been made regarding the in vitro culturing methods, also in studying the cancer microenvironment. Compared to growth in 2D systems, cancer cells grown in these 3D environments have shown different gene expression and phenotypes, highlighting the importance of these cellular interactions to oncogenesis [80]. Moreover, 3D cancer models showed high potential to simulate the tumor microenvironment and to help better understand in vivo tumor features, such as toxicity and therapeutic resistance [81,82].

### 6.2. Current Applications and Future Perspectives

Developing therapies for pancreatic diseases, such as diabetes and cancer, is hampered by a limited access to pancreatic tissue in vivo. The only curative cell therapy for type 1 diabetes mellitus is actually pancreatic islet cell transplantation [78]. However, its potential to treat many more patients is limited by several challenges. The emergence of 3D bioprinting technology from recent advances in 3D printing, biomaterials and cell biology has provided the means to overcome these challenges.

Mouse models allowed the study of environmental factors necessary for tumor growth, progression and therapeutic response [83]. Through magnetic bioprinting, 3D spheroids were obtained using a co-culture of pancreatic cancer cells and activated pancreatic fibroblasts and they have been subjected to metabolic assay [84] (Table 3). These spheroids provide an in vitro tumor model with stromal cells, which are missing in 3D models. This model can be improved and replicated for other cell types and it provides an important tool to generate functional spheroids that contain the two major cell types found in most tumor tissues, cancer cells and cancer-associated fibroblasts, which can be employed for other analysis, such as drug screening [84]. Other authors evaluated pancreatic cancer cells’ ability to form spheroids or organoids and demonstrated that this ability is influenced by the expression levels of adhesion molecules, such as β1-integrin and E-cadherin, and the interaction of β1-integrin with extracellular matrix proteins (ECM), similar to what has been demonstrated in other cancers such as hepatoma, and breast cancer [85] (Table 3).

Hakobian et al., using laser-assisted bioprinting, generated 3D pancreatic cell spheroid arrays, composed of both acinar and ductal cells; they characterized their phenotypic evolution over time and showed that they can mimic the initial stage of pancreatic ductal adenocarcinoma (PDAC) (Table 3). The analysis of internal and external factors that contribute to the formation of precursor PDAC lesions and to cancer progression can shed light on future PDAC therapy strategies [86].

Even though human models are needed to better understand the interaction between tumor and stroma, Langer et al. demonstrated that bioprinting allows for modeling patient-derived tissues in a complex microenvironment and that bioprinting- generated tissues represent the most accurate in vivo models relevant to evaluate the therapeutic response [83] (Table 3).

An important promise for diabetes treatment is islets transplantation on porous biomaterials. Some authors created a device to maintain pancreatic islets close to blood vessels in a growth factor-enriched environment which facilitates cell delivery subcutaneously [87]. In experimental studies, islets have also been implanted in the renal capsule, which also has a rich blood supply or in the omento through a fibrin scaffold. However, both technologies require invasive surgeries [88]. Further studies are needed in humans to obtain the creation of artificial pancreas for the treatment of diabetic patients. This being said, a bioprinted pancreas-on-a-chip model is possible in the near future. Bioprinting technologies may be crucial in the organization of diverse cell types and complete organ manifestation.

## 7. 3D Bioprinting of Cartilage 

### 7.1. Background

Cartilage is a flexible, aneural and avascular connective tissue with poor regenerating capacities [89]. It constitutes a very important covering tissue of articular surfaces as well as a constituent of the intervertebral discs, the intra-articular menisci and the auricle. It also has a role in providing support to some organs of the respiratory system.

Cartilage is a tissue that does not feed through the blood supply (at least not directly) but mainly through the synovial fluid, which contains nutrients and proteins capable of nourishing the cartilage tissue. On its own, cartilage is unable to repair itself following traumatic or degenerative injuries.

In the orthopedic field, there have been many attempts to regenerate cartilage (especially in joints) through the transplantation of either autologous human chondrocytes or mesenchymal cells. All these attempts, however, mainly led to the reduction of painful symptoms for the patient but not to a biologically “exact” regeneration of the native tissue. In fact, the new “generated” fibrous or fibro-cartilage tissue is very different from native cartilage. Thus, articular cartilage injuries are scarcely repaired and represent an outstanding healthcare problem, since the lesions can often progress to osteoarthritis [90]. Current strategies to treat articular cartilage defects include microfracture, mosaicplasty and cell-based techniques [91], but they have several limitations, such as the lack of effectiveness in recreating the native architecture of the tissue, the costs and the short-term resolution [92]. The fact that cartilage tissue is not a vascularized tissue, not innervated and only indirectly fed by blood makes it one of the most “easily” tackled tissues in terms of tissue regeneration, bioprinting and tissue engineering, also taking into account the difficulty of recreating neo-angiogenesis with 3D printers and, in general, with tissue engineering approaches.

Mesenchymal stem cells are shown to be the most suitable cell type to avoid the expansion of mature chondrocytes with all the negative aspects described above. The first attempt was in 1970 by Friedenstein et al., who cultured in vitro cells isolated from bone marrow, producing cartilaginous and bone sketches. In 1998 Johnstone et al. gave rise to “micromasses” with cells put in test tubes, centrifuged, and with addition of various growth factors to increase the chondrogenic possibilities of mesenchymal stem cells and to prevent their dedifferentiation (which, as reported above, occurs for in vitro expansion of mature chondrocytes).

### 7.2. Current Applications and Future Perspectives

For the lower airways defects, such as tracheobronchial malformations, the treatment options are few and they include surgical resection and subsequent end-to-end anastomosis, or the placement of splints/stents to maintain the patency of the respiratory tracts [93].

For this reason, there is a need for alternative therapeutic approaches that could bring significant benefits to patients, minimizing the side effects. There are some major hurdles in bioprinting constructs for airway defects, including the lumen patency, the mechanical stability, the necessity of long-term preclinical studies [94], the exposure to air contaminants which could impair the proper epithelium maturation [94] and the risk of granulation/inflammatory tissue formation when there is no epithelial layer covering the scaffold lumen-side [95,96,97].

Despite many important achievements in the bioprinting of joint cartilage, several major roadblocks must be overcome. Articular cartilage can be divided into four different zones which appear heterogenous for cell density, composition and mechanical properties. Reproducing this spatial organization and structural complexity through bioprinting has not been achieved yet [98]; moreover, the cartilage implant has to fully integrate with the underlying articular subchondral bone [3,99]. Further problems deal with the long-term effectiveness of engineered cartilage, the optimum of cell density related to defect size, and the concerns about laboratory procedures, the serum-based media and biomaterials containing products of animal origin [100]. 

Actually, there are not many clinical applications of mesenchymal stem cells in the regeneration of cartilage tissue in humans. Considering the existing clinical trials, (ClinicalTrials.gov) we found eight projects aimed at experimenting with the regeneration of cartilage in humans, especially in the case of osteoarthritis. Very good results were achieved: histological examinations in some trials showed cartilage regeneration, a significant decrease in pain and a return to normal post-implantation activities in 6 months.

The possibility to obtain chondrocytes starting from mesenchymal stem cells isolated from adipose tissue which, unlike stem cells from bone marrow, seem to confer a certain degree of protection to other cell types and have a lower inflammatory activity, has recently been taken into consideration [101,102].

A promising option for articular cartilage repair could be represented by the in situ bioprinting approach, which takes advantage of the possibility to precisely shape the construct directly on the defect site at the time of the surgical operation [103]. Indeed, O’Connell et al. developed a hand-held in situ printing device, called “Biopen”, endowed with two ink chambers and exploiting a coaxial extrusion-based printing mechanism [104] (Table 4). They optimized the core/shell printing method of a cell-laden bioink for this device [26], and then Biopen was tested in vivo on six sheep with chondral defects, showing some promising preliminary results [35]. In another work, Ma et al. proposed an in situ 3D printing approach, in which a robotic-assisted printing technology was applied to repair osteochondral defects in rabbits; after 12 weeks, the defect was filled with newly-formed hyaline cartilage [105] (Table 4).

To address the issue of osteochondral integration for engineered joints implants, some authors perfected biomaterials enhancing calcified cartilage differentiation. You et al. optimized a hydrogel combining alginate and broadly-distributed hydroxyapatite particles: this bioink supported the production of a mineralized matrix from embryonic chick chondrocytes [106] (Table 4). Another research group printed hMSCs in a bioink enriched with β-tricalcium phosphate, then characterized the hydrogel from the mechanical and rheological points of view, and finally assessed the expression of several chondrogenic markers, including some related to mineralization [107].

Park et al. bioprinted autologous rabbit chondrocytes and epithelial cells in a construct, alternating five layers of alginate cell-laden hydrogel and PCL (Table 4). Epithelium regenerated after three months, while mature cartilage formation was not yet achieved at one year of follow-up [108]. In another recent interesting experiment, the authors employed a PCL supportive scaffold and two hMSCs-laden hydrogels to bioprint a tracheal construct with a cartilage and a smooth muscle region, thus mimicking the structural composition and mechanical properties of human trachea [109]. 

## 8. Hurdles and Promises of Bioprinting in the Clinical Routine

Three-dimensional bioprinting is an emerging technique with a striking potential in solving those clinical issues which generally require a tissue/organ allograft. The major strengths of this technology lie in the possibility of tailoring patient-specific constructs through last-generation bioprinters and, most of all, in the potential use of autologous biological material to shape personalized constructs in regenerative medicine applications.

However, to take the step “from bench to bedside” in bioprinting is still challenging, even if the research trend of the past few years is highly focused on finding new solutions to the current problems which prevent the clinical use of this technique in tissue engineering.

In this context, the main limitations are due to the biological complexity of human tissues and organs, together with the biophysical and rheological properties of biomaterials and their actual “printability”. One of the major roadblocks to overcome resides in reproducing a functional vascular network within complex biological structures shaped in vitro: nowadays, this aspect is still an open challenge in the bioprinting field [110]. Actually, the intrinsic morphological and structural features of vasculature, which is made by a hollow and perfusable lumen enclosed by three layers with different physical properties, requires an appropriate combination of biomaterials and printing conditions which is not easy to achieve in vitro [110].

Despite this, some human tissues are certainly more easily printable than others both for their biochemical characteristics and for their actual physical consistency. Bone tissue and cartilage tissue are among the most studied tissues that have reached a good level of maturation, so it is highly probable that they can reach the patient’s bedside before the others. In the case of bone tissue, dentistry is the most interested and active medical discipline in the in vivo experimentation on humans for the implantation of 3D printed bone substitutes, with the use of biopolymers [111] functionalized with growth factors and proangiogenic factors [112]

To date, there are bone substitutes on the market obtained through the combination of bioceramic materials capable of performing an osteogenic and osteoinductive function [112].

Cartilage is another tissue whose 3D bioprinting can be tackled more easily than other tissues [113].

Indeed, cartilage tissue is not vascularized and has a general consistency favorable to in vitro bioprinting. The rheological characteristics of collagen (especially if highly concentrated) treated with molecular cross-linking techniques are favorable to 3D printing in the laboratory, and this opens the way to important future developments considering, above all, the increasing demand for cartilage substitutes due to trauma or pathologies from tissue degeneration as a result of the aging of the population.

In some specific biomedical contexts, bioprinting has reached the stage of human experimentation, as for example in skin generation, a pressing need in the dermatological sector and plastic surgery due to the complexity of the treatment of skin lesions and burns. A clinical trial is currently under way (NCT04925323) which involves the recruitment of 25 subjects from which to extract epidermal tissue to be used for the creation of 3D printed skin substitutes.

However, the clinical applicability of bioprinting should not be understood as simply oriented towards tissue regeneration. It is in fact important to remember that, to date, there are important and relevant applications of bioprinting for the in vitro study of human models.

In vitro tissue bioprinting is essential for drug screening and for the study of chemotherapeutic sensitivity of tissues in medical oncology. In the oncology field, its concrete application is to recreate the tumor microenvironment in vitro in order to study drug combinations for the treatment of cancer [114]. In this regard, numerous clinical trials are under way to bioprint patient-derived organoids for the study of the chemosensitivity of myeloma (NCT03890614) and for the study of the chemotherapy response of colorectal cancer with and without liver metastases (NCT04755907). Finally, clinical trials are also active in the study of myocardial infarction by reproducing in vitro the thromboembolic microenvironment from patient-derived biological material through the construction of 3D organoids (NCT03832153).

## 9. Conclusions

Bioprinting represents a research topic of great interest and has the potential to change the future of medical science. Its application in tissue engineering could totally subvert our concept of transplantation, eliminating all the issues linked to immunocompatibility and organ waiting lists, and ultimately creating a completely tailor-made and patient-specific tissue substitute. As aforementioned, striking results have been obtained and are continuing each day in the realm of bioprinting of heart, cartilage, skin, pancreas and many other tissues. The real challenge for the future of bioprinting is the development of printing techniques and materials which are able to provide suitable mechanical resistance before, during, and after printing as well as to simulate real anatomical tissues as much as possible. Today, the most important critical issue is the difficulty in printing soft materials such as those that make up human tissues. The poor physical consistency of these materials makes the printing process difficult and often unable to achieve complex 3D structures. Thus, the future of bioprinting hinges on the need for improvements in printing techniques and especially in the choice and formulation of proper biological materials for the construction of fabrics and organ parts. This will be only possible with a multidisciplinary approach involving complementary disciplines such as materials engineering, bioengineering and biology.

## Figures and Tables

**Figure 1 ijerph-18-10806-f001:**
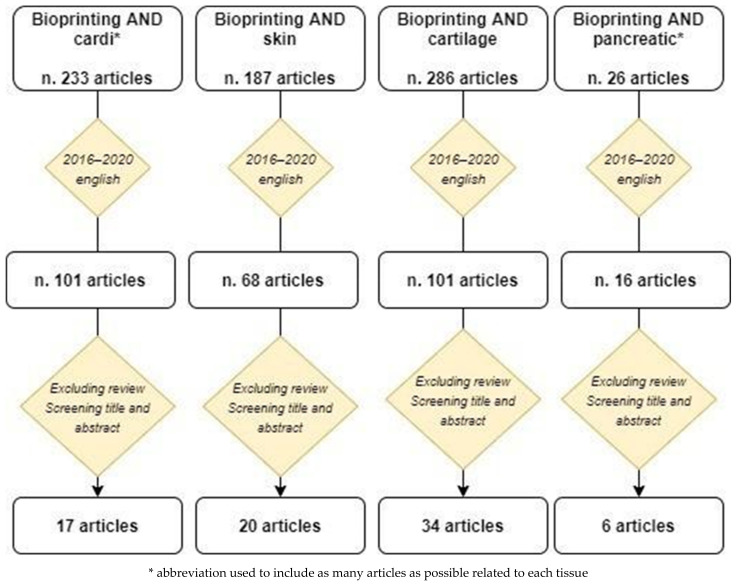
Article selection process.

**Figure 2 ijerph-18-10806-f002:**
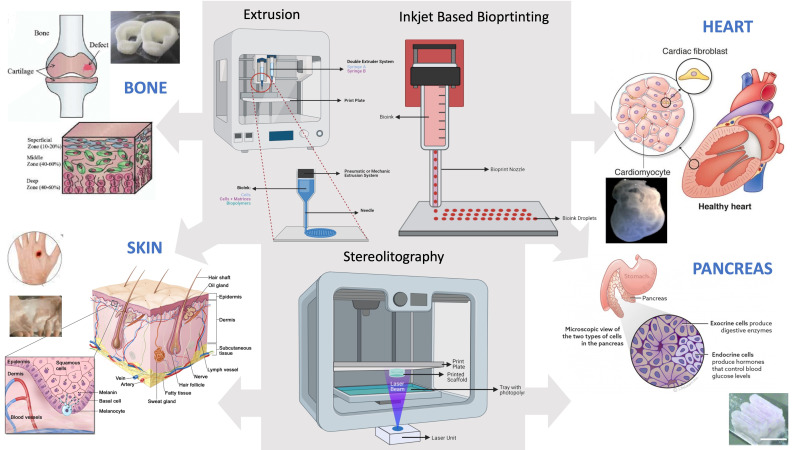
Main bioprinting techniques for different tissue-specific applications.

**Table 1 ijerph-18-10806-t001:** Remarkable achievements in 3D bioprinting of skin tissue. Articles from 2016 to 2018 and from 2019 to 2020.

Tissue	Cell Type	Biomaterial	Printed Model	Outcome	Reference
Skin					
**2016–2018**					
	Human dermal endothelial cells	Sodium alginate, chitosan, gelatin, gellan gum, collagen I (core materials), pure alginate (shell material)	Core/shell construct	- Biological, physical and mechanical characterization of the construct (with or without loaded growth factors)	Akkineni et al. 2016
	Human fibroblasts (hFB) and keratinocytes (hKC) obtained from skin biopsies of healthy donors	Plasma-derived fibrin scaffold	Skin: dermis + epidermis	- Stratification and differentiation of dermis and epidermis - Morphological similarities to native human skin - Fast bioprinting process (<35 min)	Cubo et al. 2016
	Human primary dermal fibroblasts, human primary epidermal keratinocytes	Newly-developed ECM-like bioInk	Skin model	- Proof of concept study, with the aim to set a bioprinting approach “for industrial routine application” - Viable and proliferating cells - Assessment of differentiation and stratification by histological evaluations (not fully achieved)	Rimann et al. 2016
	Amniotic fluid-derived stem (AFS) cells	Photo-cross-linkable heparin-conjugated hyaluronic acid (HA-HP) hydrogel	In situ skin graft	- Enhanced re-epithelialization and wound healing using HA-HP hydrogel + ASF - Improved vascularization of the regenerating skin tissue - ECM components secreted - In vivo study	Skardal et al. 2016
	Fibroblasts	Collagen hydrogel	Dermal construct	- Seeding of fibroblasts in a multi-layered structure of collagen hydrogel - Evaluation of the construct permeability to silica nanoparticles for drug testing: permeability rate similar to native dermal layer	Hou et al. 2017
	Human fibroblasts and human keratinocytes	Unique bioink of alginate, gelatin, fibrinogen	Skin: dermis + epidermis	- Biological/structural similarity with human normal skin - Formation of desmosomes and hemidesmosomes, epithelial maturation - Bioprinting of complex architectures (proof of concept)	Pourchet et al. 2017
	Human primary skin cells (fibroblasts and keratinocytes)	Collagen, gelatin. PCL (prevents collagen shrinkage)	Skin: dermis + epidermis	- Thick and stratified epidermis - Stretched fibroblasts - Expression of dermal and epidermal markers (collagen I, K10, involucrin) - Superior yield of KCs maturation (vs. manual seeding)	Kim et al. 2017
	Keratinocytes (KCs), melanocytes (MCs) and fibroblasts (FBs) (from three different Caucasian skin donors)	Collagen, PVP (Polyvinylpyrrolidone) polymer	Pigmented skin: bioprinting vs. manual-cast approach	- KCs and MCs markers of proliferation and differentiation, MCs anchoring at the basement membrane in the bioprinted skin - More homogeneous distribution of epidermal cells in the bioprinted pigmented skin	Ng et al. 2018
	Endothelial progenitor cells (EPCs) and adipose-derived stem cells (ASCs) added to HDF (human dermal fibroblast) and HEK (human epidermal keratinocyte)	Skin-derived extracellular matrix (S-dECM) bioink, collagen I matrix (as a control)	Skin patch	- Promotion of cellular differentiation and maturation (better properties than Collagen I) by dECM - Promotion of re-epithelialization, wound closure and neovascularization in vivo by the bioprinted pre-vascularized patch (ASCSs + EPCs in dECM)	Kim et al. 2018
	Fibroblasts, melanocytes and keratinocytes	Collagen hydrogel	Pigmented skin model (dermal + epidermal layer)	- Stratification of dermis and epidermis, and pigmentation spots	Min et al. 2018
	Human melanocytes (HEM), human keratinocytes (HaCat) and human dermal fibroblasts (HDF)	Gelatin methacrylamide (GelMA) and collagen (Col)+ tyrosinase (Ty)	Living skin model	- In vitro characterization of the bioink both from a mechanical (degradation rate, viscosity and rheological properties) and a cellular (proliferation rate, cell viability, cell migration) point of views - Promotion of melanocytes viability and proliferation by Ty - Enhanced wound healing using Ty doped bioink in in vivo studies	Shi et al. 2018
**2019–2020**					
	Fibroblasts and keratinocytes	Hydrogel (fibrinogen, collagen I, trombin)	In situ skin bioprinting	- Wound healing with re-epithelialization and vascularization, preventing scar formation (histological analysis) - Wounding and treatment of murine and porcine models with the developed in situ approach	Albanna et al. 2019
	Neonatal human dermal fibroblasts and neonatal normal human epithelial keratinocytes	Gelatin, fibrinogen, collagen, elastin (dermal hydrogel)	Skin equivalent	- Bioprinting of three different layers (dermal, basal and epidermal layers) - In vitro culture of the skin construct - Histological analysis: morphological and molecular similarities with native human skin - Structural evaluations: electrical conductivity, permeability and barrier function assessment	Derr et al. 2019
	Human fibroblasts, keratinocytes, human umbilical vein endothelial cells (HUVECs), preadipocytes	dECM-based bioinks, gelatin hydrogel. PCL transwell system (supportive mesh)	A vascularized tri-layered skin model (epidermis, dermis, and hypodermis)	- Well differentiated and stratified skin equivalent, similar to native human skin. - Presence of epidermal-dermal junction and vascular channels. - Investigation of skin stemness markers	Kim et al. 2019
	Human amniotic epithelial cells (AECs), Wharton’s jelly-derived mesenchymal stem cells (WJMSCs)	Alginate/gelatin composite hydrogels	Skin bilayered construct	- High printing precision and cell viability - Investigation on the rheological properties of the bioinks	Liu et al. 2019
	No cells	PCL and silk sericin for epidermis + CS_SA hydrogel for dermis (CS, chitosan; SA, sodium alginate)	Composite skin construct: three-dimensional skin asymmetric construct (3D_SAC)	- Morphological, mechanical and structural characterization of 3D_SAC - Analysis of 3D_SAC cytotoxic profile and antimicrobial properties - Potential application for wound dressing	Miguel et al. 2019
	Human dermal fibroblasts (HDFs)	Skin decellularized extracellular matrix (dECM)	Bioprinted 3D construct	- Derivation of a bioink from decellularized ECM - High bioactivity of the biomaterial, promoting skin development and morphogenesis	Won et al. 2019
	Neonatal human dermal fibroblasts (NHDFs), immortalized human keratinocyte cell line (HaCaT) and human umbilical vein endothelial cells (HUVECs)	Methacrylated gelatin (GelMA) and succinylated chitosan/dextran aldehyde	Prevascularized core/shell construct for wound healing	- In vitro analysis: accelerated wound healing (twofold rate compared to the control)	Turner et al. 2020
	Human-derived skin fibroblasts (hSF)	Bioink made of nanofibrillated cellulose (NFC), alginate (ALG) and carboxymethyl cellulose (CMC)	Dermal construct	- Optimization of a bioink with the desired rheological properties and the proper printability - High cell viability and proliferation	Zidaric et al. 2020

**Table 2 ijerph-18-10806-t002:** Remarkable achievements in 3D bioprinting of cardiovascular tissue. Articles from 2016 to 2018 and from 2019 to 2020.

Tissue	Cell Type	Biomaterial	Printed Model	Outcome	Reference
Heart					
**2016–2018**					
	HUVECs, neonatal rat CMs/ hiPSCs-CMs	Alginate and GelMa	Endothelialized-myocardium-on-a-chip model	- Endothelialized myocardial organoids - Spontaneous and synchronous contraction - Drug-responding model (suitable for a personalized drug screening platform)	Zhang et al. 2016
	Human adipose derived mesenchymal stem cells (HADMSC), aortic valve interstitial cells (HAVIC) and aortic valve sinus smooth muscle cells (HASSMC)	Mixture of methacrylated gelatin/polyethylene glycol diacrylate/alginate (MEGEL/PEGDA 3350/alginate)	3D-bioprinted hydrogels for cardiac valve	- Optimization of the printing conditions testing different photo- cross-linking parameters	Kang et al. 2017
	Human coronary artery endothelial cells	Sodium alginate	Cardiac constructs (different architectures)	- Analysis of the bioprinted constructs (different spatial patterns) for their mechanical/physical properties, for cell viability and for printing pattern fidelity - Ex-vivo technical evaluations	Izadifar et al. 2017
	hiPSC-CMs, FBs, ECs	Scaffold-free	Patch	- Cardiosphere fusion - Electric coordination - Vasculogenic potential	Ong et al. 2017
	hiPSC-CMs, human dermal FB and EC (HUVECs)	Scaffold-free	Tubular cardiac constructs made of cardiac spheroid	- Bioprinting of cardiac spheroids (hiPSC-CMs+ hFB + HUVECs) on a needle array platform - Formation of tubular scaffold-free constructs	Arai et al. 2018
	Rat primary cardiomyocytes	Fibrin cell-laden hydrogel, sacrificial hydrogel and a PCL supporting frame	Patch	- Spontaneous and synchronous contraction - Physiological response to cardiac drugs	Wang et al. 2018
	hCPCs, cECM	Decellularized cardiac extracellular matrix hydrogel (cECM) and gelatin methacrylate (GelMA)	Patch	- High post-printing viability - Pro-angiogenic potential in vitro - Integration and vascularization in vivo	Bejleri et al. 2018
	Bone marrow- derived human mesenchymal stem cell (hMSCs), neonatal rat CMs	Gelatin hydrogel	Patterned gelatin hydrogel 3D bioprinted grid	- Aligned F-actin fibers - Elongated appearance and patterned cell distribution along the microchannels - Myocardial commitment (cardiac markers expression) - Synchronous beating	Tijore et al. 2018
	Human coronary artery endothelial cells (HCAECs)	Carboxyl functionalized carbon nanotubes (CNTs) incorporated alginate framework and cell-laden methacrylated collagen (MeCol)	Nanoreinforced hybrid cardiac patch	- Improved mechanical, electrical and biomimetic properties - Favorable microenvironment for cell proliferation and differentiation	Izadifar et al. 2018
		**2019–2020**			
	iPSCs-derived CMs and ECs (patient-specific)	Patient-specific hydrogel (collagen/ECM)	Patient-specifically designed patch	- ECs organization - CM sarcomeric organization and contractile potential (in vitro and in vivo)	Noor et al. 2019
	hESC-CMs and cardiac FBs	Collagen	Left ventricle model, tricuspid heart valve	- Electrophysiological function - Contractility and wall expansion during contraction of the ventricle model; - High printing reliability and accuracy	Lee et al. 2019
	hiPSC-CMs, FB and EC	Scaffold-free	Patch	- Improvement of cardiac function and lifespan in vivo - Vascularization and reduction of scar area	Yeung et al. 2019
	hiPSC-CMs	Photo-cross-linkable cardiac dECM	Biomimetically patterned construct	- Printing of complex patternS - Fine microarchitectures in a mechanically tunable way through digital light processing (DLP) - High CMs viability post-printing - Maturation (expression of cardiac-specific markers) - Synchronous beating	Yu et al. 2019
	Wnt-activated hiPSC-cardiomyocytes, hiPSC-cardiomyocytes	GelMA and 0.3% (wt/vol) lithium phenyl-2 4 6-trimethylbenzoylphosphinate (LAP)	hPSC-derived cardiomyocyte mini hearts	- Mini heart-shaped construct made of hiPSC-derived cardiomyocytes (digital micromirror device bioprinting) - Wnt-activated hiPSC-cardiomyocytes as pacemaker-like cells, initiating the electrical activity of the other hiPSC-cardiomyocytes	Ren et al. 2019
	hiPSC-CM and normal human cardiac fibroblasts (NHCFs) coated with fibronectin and gelatin	Fibrinogen and hyaluronic acid	Layer-by-layer heart construct hiPSC-CM-derived	- High cell viability and density - Synchronous beating - Expression of cardiac markers (F-actin and cardiac troponin T)	Chikae et al. 2019
	iPSCs-derived CMs, ECs	Photo-cross-linkable bioink of ECM proteins and GelMa	Chambered cardiac pump	- Thick walls and electromechanical function - CMs differentiation, functional (6 weeks)	Kupfer et al. 2020

**Table 3 ijerph-18-10806-t003:** Remarkable achievements in 3D bioprinting of pancreatic tissue.

Tissue	Cell Type	Biomaterial	Printed Model	Outcome	Reference
Pancreas					
	Human pancreatic islet cells	Polylactic acid functionalized with growth factor-enriched platelet gel	3D printed construct	- Adequate and prompt vascularization of the graft - Vascularization enhancement by functionalized scaffold’s ability to dispense proangiogenic factors, such as VEGF, also known to increase islet viability and function - Avoiding surgical retrieval due to the transcutaneous refillability of the device - Significant advantage in the case of children with diabetes	Farina et al. 2017
	Pancreatic cancer cells (Patu8902) and activated pancreatic fibroblast cells (PS1)	Nanoshuttle (NS) composed of iron oxide, poly L-lysine and gold nanoparticles	In vitro pancreatic tumor model	- 3D spheroids based on pancreatic cancer cells and activated pancreatic fibroblasts (400–600 μm in diameter) obtained by magnetic bioprinting - Quick, easily adaptable and consistent, able to resemble the in vivo tumor microenvironment, comparatively inexpensive method - Efficient tool for the tumor biology and drug screening studies	Noel et al. 2017
	Human colorectal adenocarcinoma cell line HT-29, human pancreatic epithelial carcinoma cell line PANC-1	Nanoshuttle (NS) composed of iron oxide, poly L-lysine and gold nanoparticles	Primary pancreatic organoid tumor models	- Development of an efficient high-throughput screening (HTS) method for the production of organoids, by combining the use of a cell-repellent surface with a magnetic force-based bioprinting technology - Validation by investigating the anticancer agents’ effects against four patient-derived pancreatic cancer KRAS mutant-associated primary cells - Cytotoxicity pilot screen of ~3300 approved drugs - Readily applicability to support large-scale clinical drug screening on ex vivo 3D tumor models directly harvested from patients	Hou et al. 2018
	Pancreatic cancer cell lines, i.e., MIA PaCa-2 and PANC-1	NanoShuttle nanoparticles (Nano3D Biosciences Inc., Houston, TX, USA)	Spheroids from MIA PaCa-2 and PANC-1 cells, mixed with human fibroblasts in a ratio of 1:1, and incubated with NanoShuttle nanoparticles	- The greatest effect on tumor spheroid growth in both cell lines with the combinations of ICPD47, inhibitor of Hsp90 (heat shock protein 90) with the antimetabolites gemcitabine (GEM) and 5-fluorouracil (5-FU) in a ratio of 1:5 - Significant dropping of the EC50 value in PANC-1 cell line from 4.04 ± 0.046 to 1.68 ± 0.004 μM, in the case of the ICPD47 combination with mild hyperthermia - Synergistic action of the Hsp90 inhibitors, i.e., ICPD47 and ICPD62, with GEM, 5-FU and the topoisomerase inhibitor doxorubicin (DOX), under the same conditions	Daunys et al. 2019
	Human primary pancreatic stellate cells (PSCs), human umbilical vein endothelial cells (HUVECs), HMF, subcutaneous preadipocytes(SPA), and MCF-7 cells	Alginate-containing hydrogel		- Application of 3D bioprinting to generate multicellular, architecturally defined, scaffold-free tissue models of human tumors - Use of Organovo’s Novogen MMX Bioprinter Platform to print structures composed of a cancer cell core surrounded by several stromal cell types	Langer et al. 2019
	AR42J-B-13 rat acinar cell line	Methacrylated gelatin (GELMA)	Laser-assisted bioprinted 3D pancreatic cell spheroid arrays	- Suitability of the laser-assisted bioprinting to generate cellular spheroid arrays with high control over cell number deposition and spatial resolution - Replication of the initial stages of the pancreatic ductal adenocarcinoma by means of the bioprinted miniaturized spheroid-based array model, composed of both acinar and ductal cells - Utility of the model to study the internal and external factors that contribute to the precursor PDAC lesions formation and to cancer progression	Hakobyan et al. 2020

**Table 4 ijerph-18-10806-t004:** Remarkable achievements in 3D bioprinting of cartilage tissue. Articles from 2016 to 2018 and from 2019 to 2020.

Tissue	Cell Type	Biomaterial	Printed Model	Outcome	Reference
Cartilage					
**2016–2018**					
	Rabbit ear chondrocytes	PCL, gelatin, fibrinogen, HA (hyaluronic acid)	PCL and chondrocytes laden scaffold	- Ear cartilage reconstruction	Kang HW et al. 2016
	hMSC	Nanocrystalline hydroxyapatite	Scaffold	- Influence of the scaffold pore distribution on mechanical and biological properties - Successful osteogenic and chondrogenic manipulation in the scaffolds	Nowicki et al. 2016
	Human mesenchymal stem cell	2D nanosilicate reinforced kappa-carrageenan (κCA) hydrogels	Hydrogel scaffold	- High cell viability in the case of cells encapsulated within shear-thinning nanocomposite such as nanosilicate reinforced kappa-carrageenan - Maintenance of cells’ round shaped morphology over a week indicating that the developed material can be used for soft tissue regeneration	Thakur et al. 2016
	Human mesenchymal stem cells (hMSCs)	Poly(ethylene) glycol diacrylate (PEGDA)/acrylated peptides/I-2959 photoinitiator	bioprinted 3D construct	- No or minimal immune response after 3D printed tissue construct subcutaneous implantation in mice - Enhanced chondrogenesis in the case of nuclear receptor subfamily 2 group F member 2 (NR2F2) overexpressed MSCs - Remarkably higher proteoglycan production in NR2F2 overexpressed MSCs - Significantly enhanced compressive modulus in the implanted scaffold with NR2F2 overexpressed MSCs, mainly due to the accumulated cartilage matrix in the scaffold	Gao et al. 2016
	Human adipose stem cells	Gelatin–methacrylamide/hyaluronic acid–methacrylate (GelMa/HAMa) hydrogel	Hand-made 3D Scaffold	- In vitro high viability (>97%) of human adipose stem cells one week after Biopen printing in GelMa/HAMa hydrogels - Potential use of the developed hand-held biofabrication tool (Biopen) for 3D bioprinting during the surgical process - Ability to directly control the deposition of regenerative scaffolds with or without the presence of live cells during the surgery - Possibility of surgical sculpting of the substitute tissue to achieve the desired structure - Increased surgical dexterity - Small dimensions and easy transportation in/out of the surgical field - Easiness of sterilization and sterile condition maintenance - Biopen has a low cost	O’Connell et al. 2016
	hMSCs and human nasal chondrocytes	Nanofibrillated cellulose and alginate	5 × 5 × 1.2 m biological construct	- Good proliferation of chondrocytes in the scaffold after 60 days - Production of glycosaminoglycan and type 2 collagen.	Apelgren P. et al. 2017
	iPSCs	Nanofibrillated cellulose and alginate (NFC/A) or hyaluronic acid (NFC/HA)	Scaffold	- Hyaline-like cartilaginous tissue with collagen type II expression in 3D-bioprinted NFC/A	Nguyen D. et al. 2017
	Chondrocytes	Methacrylated hyaluronic acid (HAMA) + methacrylated poly[N-(2-hydroxypropyl) methacrylamide mono/dilactate] (pHPMA-lac)/polyethylene glycol (PEG) + polycaprolactone (PCL)	Scaffold	- Increase of glycosaminoglycan (GAG) and type II collagen contents by Hama - HPMA-lac-PEG hydrogels optimal for cartilage-like tissue formation - Young’s moduli of hydrogel co-printed with PCL similar to cartilage	Mouser et al. 2017
	Human and equine mesenchymal stem cells (hMSCs)	Hyaluronic acid/poly(glycidol) and poly(ε-caprolactone)	Bioprinted 3D construct	- Chondrogenic differentiation stimulation by the hyaluronic acid included in the hydrogel - Possibility to print scaffolds with suitable mechanical characteristics and supported by PCL strands by means of the double printing technique	Stichler et al. 2017
	Human embryonic kidney (HEK) cells and ovine mesenchymal stem cells (oMSCs)	8:1 v:v mixture of ULGT-agarose solution to Fmoc-dipeptide solution, with or without collagen	High-resolution patterned 3D cellular constructs	- Tissue relevant densities (10^7^ cells mL^−1^) and high droplet resolution of 1 nL for printing HEK and oMSCs cells - High resolution of 3D geometries, including an arborized cell junction, a diagonal-plane junction and an osteochondral interface, with features of ≤200 μm - Proliferation of HEK cells within the printed structures under culture conditions - Ability of the printed oMSCs to be differentiated towards the chondrogenic phenotype to generate cartilage-like structures containing type II collagen	Graham et al. 2017
	MSCs	GelMA + PEGDA + TGF-β1 embedded in nanospheres	Stereolithography Scaffold	- Improvement of the printing resolution by PEGDA addition in GelMa - The highest cell viability and proliferation rate in the case of 5%/10% (PEGDA/GelMA) hydrogel - Chondrogenic differentiation improvement byTGF-β1	Zhu et al. 2018
	hADSCs	GelMa + hyaluronic acid methacrylate	Scaffold with Biopen	- Human hyaline-like cartilage formation	Onofrillo et al. 2018
	hMSCs	Poly (l-lactide-co-caprolactone) + poly (lactic-co-glycolic acid) + Aggrecan	Scaffold	- Expression of type II collagen 10 times higher compared to standard micro fracture approach	Guo et al. 2018
	Human chondrocyte	Gelatin methacryloyl bioink	Three-dimensional disks	- Good printability of gelatin methacryloyl with high print resolution and high cell viability in the case of 10%*w*/*v* concentration - High potentiality to extend the developed technique to tissue engineering of soft tissue other than cartilage	Gu et al. 2018
	Mesenchymal stem cells (MSCs)	Gelatin methacrylamide (GelMa) and hyaluronic acid methacrylate (HAMA) hydrogel	Hand-made 3D Scaffold	- Possibility to obtain an in situ geometric control and to lay down multilayer biological materials, using the Biopen - Possible regeneration of full-thickness chondral defects in a mouse model - Hand-made scaffolds have macroscopic and microscopic characteristics comparable to the 3D bioprinted scaffolds - Hyaline cartilage formation with tissue regeneration, demonstrated by a columnar arrangement of chondrocytes, evidenced by histological exams	Di Bella et al. 2018
			**2019–2020**		
	Human chondrocytes	Polylactic acid	Reticular layered scaffold	- Apoptosis, proliferation and metabolic activity in an innovative Volume-by-volume 3D printing technique	Baena J.M. et al. 2019
	BM-MSCs	Scaffold-free	Spheroid	- Increase of the expression of osteogenesis and chondrogenesis associated genes between weeks 2 and 6 - Bone-cartilage interaction - Intramembranous ossification - Metaplastic transformation of cartilage into bone	Breathwaite E.K. et al. 2019
	Human mesenchymal stem cells (hMSCs)	Polycaprolactone (PCL)	Bioprinted 3D tracheal shape construct	- Two distinct scaffold designs for cartilage and smooth muscle tissue regeneration with cartilage native mechanical properties in the final 3D bioprinted construct - MSCs inclusion in two different hydrogels containing different growth factors to induce differentiation into chondrocytes and smooth muscle cells - Cartilage and smooth muscle formation in the case of both cell lines after two weeks	Dongxu et al. 2019
	hMSCs and hACs (human artificial chromosomes)	GelMa + CS-AEMA (chondroitin sulfate amino ethyl methacrylate) Hyaluronic acid (HC) + TCP (tricalcium phosphate) microparticles	Reticular layered scaffold	- Regeneration of osteochondral surface with multiple material extrusion based on microfluidic channel system - Suitability of the developed system to produce tissue with transient properties in the same biological construct	Idaszek J. et al. 2019
	hMSC	PCL	Scaffold	- Final construct elastic modulus and yield stress comparable to that of native tracheal tissue - Cartilage formation in the cultured regions	Ke et al. 2019
	ATDC5 cells	Oxidized hyaluronate (OHA), oxidized hyaluronate (OHA), oxidized hyaluronate (OHA)	Scaffold	- Critical influence on the polymer concentration and molecular weight of polymers on the 3D printability and mechanical properties of construct - No significant influence on the ATDC5 cells encapsulated in the hydrogel by the 3D printing process	Kim et al. 2019
	BM-MSCs	β-tricalcium phosphate (TCP)	3D biomimetic hydrogel scaffold	- High degree of bm-MSCs, viability - Deposition of Collagen1, Collagen2 and Collagen10 protein in 3D printed scaffolds	Kosik-Kozioł et al. 2019
	hADSCs	hydroxybutyl chitosan (HBC) + oxidized chondroitin sulfate (OCS) Hydrogel	Macroporous hydrogel scaffold	- Good viability for hADSCs in HBC/OCS hydrogel - Inhibition of acute inflammatory response in 7d by the hydrogel	Li et al. 2019
	Human cartilage cells + human fibroblasts + human umbilical vein endothelial cells + human mesenchymal stem cells	Culture medium	Spheroid	- Proliferation of tracheal epithelium and capillaries	Machino et al. 2019
	MSCs	Cartilage extracellular matrix (cECM)-functionalized alginate bioink	Scaffold	- In vitro chondrogenesis - High cell viability - The highest level of Collagen2 and ACAN expression in the case of bioink with high cECM concentration	Rathan et al. 2019
	hMSCs	PEGDA	GDF5-conjugated BMSC laden scaffold	- Scaffold 3D printing with PCL, BMSCs and GDF5 for cartilage tissue regeneration	Sun et al. 2019
	No cells used	CTGF (connective tissue growth factor) + TGFβ3 (transforming growth factor beta 3) + BMP2 (Bone Morphogenetic protein 2). All these growth factors are encapsulated in PLGA—poly(lactic-co-glycolic acid) scaffold	Thin membrane-like scaffold	- In vitro differentiation of mesenchymal progenitor cells with formation of tendon-like, cartilage-like and bone-like tissue - Promotion of the endogenous tendon progenitor cells recruitment by the developed scaffolds with formation of strong fibrocartilaginous interface - High potentiality of improving outcomes for rotator cuff repair with 3D bioprinting of biological scaffold	Tarafder S et al. 2019
	Human auricular chondrocytes (hACs)	poly(2-ethyl-2-oxazoline) (PEOXA)-peptide conjugates + sortase A (SA) + alginate + cellulose nanofibrils (CNF)	Scaffold	- Cell viability more than 90% in the case of PEOXA-Alg-CNF with hACs	Trachsel et al. 2019
	ADSCs	Alginate support	Spheroid	- Cartilage- and bone-specific gene and protein expressions in the case of differentiated ADSC spheroids - Tightly integrated organization between chondrogenic and osteogenic zone for the osteochondral interface (OC)	Ayan et al. 2020
	Ovine fetal chondrocytes	Collagen I + fibrin glue	Scaffold	- In vitro cartilage formation supported by Collagen I - Limitation due to the construct shrinking	Dasargyri et al. 2020
	hADSCs	10%GelMa/2%HAMa Hydrogel	Core/shell bioscaffold	- High cell viability - Adequate mechanical properties for articular cartilage regeneration and repair	Duchi et al. 2020
	hMSC	Calcium phosphate cement (CPC) + alginate-methylcellulose (algMC)	Scaffold	- hMSC differentiation and ECM components production after 3 weeks	Kilian et al. 2020
	Human adipose tissue-derived mesenchymal stem cells (hADMSCs)	Photo-cross-linkable alginate + gelatin and chondroitin sulfate + graphene oxide	Scaffold	- Higher in vitro proliferation of hADMSCs in bioconjugated nanocomposite inks compared to pure alginate - hADMSCs adhesion on scaffold at 7 days - Ability of the nanocomposite hydrogels to guide the cell proliferation along the direction of 3D printed threads	Olate Moya et al. 2020
	ADSCs	GelMA + PEGDA coated with lysine-rosette nanotubes (RNTK)	Lysine-functionalized rosette nanotubes scaffold	- Chondrogenic differentiation - Collagen II and glycosaminoglycan synthesis - Gene expression of collagen II α1, SOX 9, and aggrecan in the ADSCs	Zohu et al. 2020

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
