# Peer review of "Bioprinting Technology in Skin, Heart, Pancreas and Cartilage Tissues: Progress and Challenges in Clinical Practice"

_ijerph, 2021, doi:10.3390/ijerph182010806_

Round 1

Reviewer 1 Report

In this manuscript, the authors have reviewed advances in the realization of skin, cardiac, pancreas, and cartilage bioprinted substitutes, then briefly discussed the current challenges and future prospects of 3D bioprinting in the design and realization of complex tissues. While the subject is rather wide and difficult to be covered, the authors' effort was evident throughout the manuscript to extend details as much as possible. However, there are some major comments for the authors:

  1. The quality of Figure 1 is very poor. It would be better if it could be redrawn to improve clarity.
  2. On page 6, line 181, the authors mentioned “ together with a high cost of production ”, can this disadvantage be overcome for bioprinting?
  3. Table 1, Table 2, and Table 4 are currently rich in content, but not well presented. It would be better if they could be classified.
  4. Lack of reference on page 13, line282-284
  5. In section 7, the authors mentioned “Many authors also attempted to bioprint functional tracheal constructs” and related research was quoted. But it was only a statement of the research content, without corresponding comments and perspectives.

Reviewer 2 Report

The review manuscript from Piazza et al. shines light on the progress of bioprinted tissues in clinal use and the related challenges. The strength of the manuscript comes from its comprehensive summary of the so far published bioprinted tissues and their benefits and limitations. However, the knowledge related to the clinal practice is scattered throughout the text without any clear informative summary. As the review wishes to focus on the clinical practices, the authors should add a chapter where the general needs and challenges coming from the clinical practice are discussed. Especially, as the authors claim that we are every day closer to the clinical application of bioprinting, it would be beneficial to elaborate in more detail, what specific characteristics of the latest achievements make the clinical use more possible. Numerous readers come from outside of clinical practice and for them the expert's view of what is required of materials or a printing technique in clinical use would be highly beneficial.  In the manuscript, the final conclusions are too general and also lack the focus on the clinical practice. The best review articles provide the readers with new ideas and an educated forecast of the future in the field. This would strengthen the current manuscript as well. What are the future steps to be taken to bring the new bioprinting materials and bioprinted tissues to the clinical use? What should we improve in our current materials and bioprinting techniques before this can happen? Such reflection could contribute to moving the bioprinting field forward in the clinical practice.

Reviewer 3 Report

This review provides an overview of some of the latest and remarkable achievements in the realization of skin, cardiac, pancreas and cartilage bioprinted substitutes, with the purpose of underlying for each tissue the critical shortcomings that currently keep this technique far from translational reality.   As such, The figures and tables in the review are clear, and the logic of the full text is clear and coherent, However, there are several questions need to be addressed,

1) The overall quality of the English language needs to be improved. 2) How is the author's review paper different from other Review papers in this field?   3) How did the author search for these references? Does it basically contain research publications in the field over the past ten years to make the review more comprehensive?

Once the above concerns are fully addressed, the manuscript could be accepted for publication in this journal.
